# Preparation of Modified Chitosan Microsphere-Supported Copper Catalysts for the Borylation of α,β-Unsaturated Compounds

**DOI:** 10.3390/polym11091417

**Published:** 2019-08-28

**Authors:** Wei Wang, Zufeng Xiao, Chaofan Huang, Kewang Zheng, Yin Luo, Yumin Dong, Zitong Shen, Wei Li, Caiqin Qin

**Affiliations:** 1School of Chemistry and Materials Science, Hubei Engineering University, Xiaogan 432000, China; 2Key Laboratory of Biological Resources and Environmental Biotechnology, Wuhan University, Wuhan 430000, China

**Keywords:** chitosan microspheres, copper, organoboron compounds

## Abstract

Chitosan microspheres modified by 2-pyridinecarboxaldehyde were prepared and used in the construction of a heterogeneous catalyst loaded with nano-Cu prepared by a reduction reaction. The chemical structure of the catalyst was investigated by Fourier Transform Infrared Spectroscopy (FT-IR), Scanning Electron Microscopy (SEM), Transmission Electron Microscopy (TEM), and X-ray Photoelectron Spectroscopy (XPS). Under mild conditions, such as no ligand at room temperature, the catalyst was successfully applied to catalyze the borylation of α,β-unsaturated receptors in a water-methanol medium, yielding 17%–100% of the corresponding β-hydroxy product. Even after repeated use five times, the catalyst still exhibited excellent catalytic activity.

## 1. Introduction

The development of heterogeneous catalysts has become a core approach to green, sustainable chemistry due to their ease of recycling, reusability, and environmental friendliness in comparison to homogenous catalysts [1,2]. The immobilization of metal particles on a polymeric carrier is an effective method for preparing heterogeneous catalysts. Chitosan is a preferred polymer carrier for dispersing nanoparticles because of its excellent physical properties, such as being insoluble in most solvents, and can be conveniently prepared as film, fibers, and microspheres [3,4]. Additionally, chitosan can be easily derivatized due to the large amount of embedded active hydroxyl groups and amino groups which are directly used as organic catalysts [5,6]. These functional groups can also be modified into schiff bases [7,8], to enhance the ability to chelate ions or elementary substances. Most importantly, chitosan polymers can also be easily recycled and reused. There are several reports on the application of chitosan-supported metal catalysts, which include the reduction of nitrobenzene [9,10], epoxidation of olefins [11], Suzuki cross-coupling reactions [12], Heck cross-coupling reactions [13], Knoevenagel condensation [14], and Michael addition [15,16].

Transition metals have been widely applied in the catalysis of many organic reactions as they have high regio- and chemo-selectivity, and are increasingly used in modern organic synthesis [17,18]. Compared to noble metals such as Pd [19,20] and Ni [21] copper, their salts are more suitable for green and sustainable chemistry due to their abundant sources, low cost, low toxicity, and high stability [22,23]. The active component of copper used as a catalyst was first applied to the Ullmann coupling reaction [24] and then gradually applied to Sonogashira coupling [25], C–N cross-coupling [26], arylation of the aryl C–H bond [27], and C–H functionalization [28,29].

Since the C–B bond of organo-boron compounds can maintain an absolute configuration after the conversion of functional groups such as C–O, C–N, and C–C bonds [30], organo-boron compounds are widely used as intermediates in organic synthesis [31] and in the pharmaceutical industry [32]. In 2000, Hosomi et al. [33] first reported the conjugated borylation of α,β-unsaturated ketones catalyzed by the Cu^I^ catalytic system, in which cuprous sulfonate was used as the catalyst, tributylphosphine as a ligand, pinacol diboron as a nucleophile, and Dimethyl sulfoxide (DMF) as a solvent. Chain and cyclic α,β-unsaturated ketone borylation reactions have been further investigated. Compared to the use of Cu^I^ in the organic phase, the catalytic borylation reaction with Cu^II^ or Cu^0^ in the aqueous phase is simpler, greener, and safer. In 2010, Yun and co-workers [34] used Cu^II^ oxide as a catalyst to study the boron addition reaction of α,β-unsaturated aldehyde in a mixed solvent of water and tetrahydrofuran, which was the first example performed in an aqueous phase. In 2013, Kobayashi et al. [35] used pure water as a solvent and creatively used Cu^0^ powder to achieve asymmetric borylation.

The use of chitosan as a carrier of Cu^II^ or Cu^0^ catalysts in the boron addition reaction can overcome the disadvantages that the catalyst is not easily recyclable. Our group has successfully prepared two highly active and easily recoverable catalysts, chitosan@Cu(OH)_2_ [36] and chitosan/poly(vinyl alcohol)@Cu [37], which have been used to catalyze the borylation of α,β-unsaturated receptors at room temperature and in aqueous media.

In this work, we continued to explore the application of chitosan-supported Cu^0^ catalysts in boron addition reactions. We prepared 2-pyridineformaldehyde modified chitosan microspheres (CSM) by an emulsion reaction, and then obtained the corresponding catalytic materials (CSM@ Cu^0^) using sodium borohydride. In the sample vial, 1 mol% Cu catalyst exhibited good catalytic activity in the borylation reaction of different unsaturated acceptors in a water/methanol solvent, even after repeated use five times.

## 2. Materials and Methods 

### 2.1. Materials

Chitosan (CS, deacetylation degree 95%, viscosity > 400 mpa·s) was purchased from Nine-Dinn Chemistry (Shanghai) Co. Ltd., China. 2-Pyridinecarboxaldehyde, Bis(pinacolato)diboron (purity >98%), Copper sulfate pentahydrate (purity > 99%), Sodium borohydride (purity > 98%), Sodium perborate tetrahydrate (purity > 97%), Epichlorohydrin, Tween-80, Span-80, Isopropanol, Methanol, Ethyl acetate, Petroleum ether, and all α,β-unsaturated compounds were purchased from Aladdin Chemical Co. Ltd., Shanghai, China. Copper standard solutions (100 μg/mL) were purchased from the NCS testing technology Co. Beijing, Ltd., China. Distilled water was purchased from the A. S. Watson group Ltd., Xiaogan, China.

### 2.2. Synthesis of Chitosan Microspheres

Chitosan microspheres were prepared according to the previously reported procedure [38]. A total of 20 mL of 0.02 g/mL chitosan-acetic acid solution (acetic acid solution, 2%, *v*/*v*) was used as the aqueous phase. Additionally, 100 mL of isooctane containing 2 mL of tween-80 and 1 mL of span-80 was used as an oil phase. Then, the aqueous phase (10 mL) was slowly added dropwise to the oil phase (50 mL), emulsified, and stirred to form a W/O emulsion. Afterwards, a solution of 2-pyridinecarboxaldehyde in acetone was added, and the mixture reacted at room temperature overnight. Finally, modified chitosan microspheres 1 (CSM) were collected by centrifugation and washed repeatedly with petroleum ether, isopropanol, and ionized water before lyophilization.

Microspheres **1** (50 mg) were treated with 20 mL of 40 mg/L copper sulfate solution. After adsorbing copper (II), they were filtered, washed with water, and dried under vacuum to give a solid **2**(CSM@Cu^II^). Finally, the Cu^II^ supported on the chitosan microspheres was reduced by NaBH_4_ to obtain the final Cu^0^ catalyst **3** (CSM@Cu^0^). The preparation process of catalyst **3** is illustrated in Figure 1.

### 2.3. Analytical Methods

CHN Elemental Analyzer (Thermo, Waltham, MA, USA) and Scanning Electron Microscopy (SEM) (JEOL, JSM-6510, Japan) were used to measure the morphology and sizes of the modified chitosan microspheres. The Fourier Transform Infrared Spectroscopy (FTIR) spectra of the microspheres were obtained using a Nicolet iS5 Spectrophotometer (Thermo, USA) to investigate possible interactions between the microspheres and the nano-copper. The thermal stability of the microspheres was analyzed using a Thermal Gravimetric Analyzer (TGA, Netzsch, STA449F5) at a heating rate of 10 °C/min under N_2_ flow (40 mL/min). Copper morphology and lattice spacing were analyzed by Transmission Electron Microscopy (TEM, JEOL-2100F, Japan). X-ray Photoelectron Spectroscopy (XPS, ESCALAB 250xi, Thermo, USA) and X-ray Diffractometry (XRD, S2, RIGAKU, Japan) were used to obtain the elemental valence of the catalyst copper. The concentration of Cu in the catalyst was determined by Inductively coupled plasma atomic emission spectrometry (ICP-OES) (IRIS Intrepid II, Thermo). Purification of the product was carried out using column chromatography (silica gel, 200-300 mesh), and the structure was examined by Nuclear Magnetic Resonance (Bruker Avance III 400 Hz, Rheinstetten, Germany).

### 2.4. Adsorption Kinetics Experiments of Cu ^II^

In the adsorption kinetics experiments, 50 mg of chitosan microspheres **1** and 50 mL aqueous solution containing different Cu^2+^ contents were vibrated at a rate of 220 r/min at 25 °C. After a period of time, samples were taken and the content of copper ion was tested by ICP-OES. The percentage adsorption and adsorption capacity were calculated using the Equations (1) and (2): (1)$$\mathrm{Adsorption}\%=\frac{Co-Ci}{Co}\;100\%$$
(2)$$qi=\frac{\left(Co-Ci\right)\cdot V}{m}$$
where *Co* (mg/L) is the initial concentration of Cu^2+^, *Ci* represents the concentration of Cu^2+^ at time *t_i_*, *m* is the input amount of chitosan microspheres (g), and *V* denotes the volume of the solution.

The adsorption process of Cu^2+^ on the adsorbent is usually expressed as a Pseudo-first-order Equation (3) and a Pseudo-second-order Equation (4).
(3)$$\mathrm{lg}\left(q_{e}-q_{t}\right)=\mathrm{lg}q_{e}-\frac{k_{1}}{2.303}t$$
(4)$$\frac{t}{q_{t}}=\frac{1}{k_{2}q_{e}^{2}}+\frac{t}{q_{e}}$$
where *t* is the contact time; *q*_t_ and *q*_e_ represent the adsorption capacity at time *t* and the equilibrium adsorption capacity, respectively; and *k*_1_ and *k*_2_ are rate constants.

## 3. Results and Discussion

### 3.1. Characterization of the Modified Chitosan Microspheres

The morphology and diameter of the modified chitosan microspheres are shown in Figure 2. The chitosan modified by 2-Pyridinecarboxaldehyde was mostly spherical, and the diameter of the particles was not uniform. Most of the microspheres had sizes ranging between 10 and 60 μm (Figure 2c). The content of N in the chitosan microspheres before and after modification was tested using the CHN elemental analyzer. The results showed that approximately 5% of the nitrogen sites in the raw materials were acetylated, and approximately 50% of the remaining amino groups had been functionalized with pyridine-2-carbaldehyde (Appendix A). 

### 3.2. Dynamic Adsorption of the Modified Chitosan Microspheres

The dynamic adsorption of chitosan microspheres was demonstrated by the adsorption properties of copper ions. The copper (II) concentration gradient was set to investigate the adsorption capacity of the chitosan microspheres (Table 1, Entry 1-5). The effect of pH on the adsorption performance of the microspheres was first studied. When the pH of the solution was 2, the adsorption capacity of chitosan (*q*_i_) was only 2.7 mg/g. As the pH of the solution increased, the adsorption capacity of chitosan increased sharply (Table 1, Entry 5 and 16–18). When the pH of the solution was 5, the adsorption capacity (*q*_i_) increased to 99.94 mg/g. This phenomenon might be due to competitive adsorption between hydrogen and copper ions in the solution [39]. When the acidity of the solution was strong, NH_2_ or –NH– was easily protonized and could not coordinate Cu^2+^. This phenomenon may also be caused by the swelling ability of modified chitosan microspheres (CMS) at different pH values, which affects the adsorption capacity for copper ions (Appendix A). Next, the dynamic adsorption of Cu^2+^ by chitosan microspheres was studied at room temperature and at an initial Cu^2+^ concentration of 100 mg/L. The entire adsorption process could be completed in 120 min, with an adsorption rate as high as 97.8% (Table 1, Entry 6–15). These dynamic data were fitted using a pseudo-first-order reaction model and a pseudo-secondary reaction model to determine the rate determination step. As shown in Figure 3a and Figure 4b, the experimental data are more in line with the Langmuir model, which means that the diffusion of copper ions had a great influence on the adsorption process, which was the determining step of the entire adsorption process. Depending on the experimental data fitting model, the maximum adsorption capacity of chitosan microspheres was up to 99.9 mg/g. Referring to a copper (II) concentration of 40.75 mg/L (Table 1, Entry 2), we prepared nano-Cu^0^ supported on chitosan microspheres by sodium borohydride reduction, and tested the related properties of the catalyst.

### 3.3. Characterization of Supported Cu^0^ Catalysts.

#### 3.3.1. Thermo-Gravimetric Analyzes

The thermal degradation behaviors of chitosan (CS), modified chitosan microspheres (CSM), Cu^2+^-loaded modified chitosan microspheres (CSM@ Cu^II^), and Cu-loaded modified chitosan microspheres (CSM@ Cu^0^) are shown in in Figure 4. It was observed that the thermal degradation of CS is a one-step reaction [40]. There was a small weight loss process (about 12%) before 100 °C, which was caused by the water of crystallization in the material. The strong degradation of chitosan occurred at 250–350 °C, and the weight loss was around 40%. The degradation was slow at 350–600 °C, the weight loss was about 15%, and the thermal degradation ended at 700 °C. The TG curve of thermal degradation of 2-pyridinecarboxaldehyde-modified chitosan microspheres (CSM) was similar to that of CS. There was a small weight loss process (about 10%) before 100 °C, followed by the strong degradation of chitosan at 230 °C. The low thermal degradation temperature indicated that the modified chitosan had a lower crystallinity than the raw chitosan. The TG curves of thermal degradation of CSM@Cu^II^ and CSM@Cu^0^ were also similar to CSM. With the introduction of Cu^2+^ or Cu, the thermal degradation stage of chitosan was also weakened. These results indicated that CSM@ Cu^II^ and CSM@ Cu^0^ could be used as catalytic materials at a temperature below 200 °C.

#### 3.3.2. FTIR Characterization

FTIR spectroscopy could be used to confirm the structure of modified chitosan and to analyze interactions between chitosan microspheres and Cu^2+^ or Cu^0^ [41,42,43]. There were several characteristic peaks of CS, as shown in Figure 5 [44]. The peak of 3286 cm^−1^ belonged to O–H, the peak of 2920 and 2877 cm^−1^ belonged to C–H, the peak of 1667 cm^−1^ belonged to amide I (C=O stretching in NH_2_–C=O), the peak of 1592 cm^−1^ belonged to N–H (–NH_2_ bending), and the peak near 1078 cm^−1^ was the characteristic absorption frequencies of β-d-pyranoside in chitosan. Compared with CS, CSM showed new bands and changes in the infrared spectrum. As shown in Figure 5b, the peak near 3270 cm^−1^ became broader, indicating an increase in free OH groups in the modified chitosan. A new band near 1651 cm^−1^ appeared, which can be attributed to the Schiff base group (C=N). The peaks of 1569, 1472, and 776 cm^−1^ were attributed to the vibration absorption peak of the pyridine ring. These spectral data indicated that the chitosan microspheres modified by 2-pyridinecarboxaldehyde had been successfully prepared. After Cu^2+^ was adsorbed by CSM, the C=N peak shifted to about 19 cm^−1^ in the lower wavenumber direction, and became 1632 cm^−1^. At the same time, the corresponding band on the pyridine ring also moved towards the lower wavenumber. These results indicated that the N atom of the imino group and the N atom of the pyridine ring were successfully chelated with the metal copper ions.

#### 3.3.3. XPS and XRD Characterization

The XPS spectra of the modified chitosan microsphere-loaded Cu^2+^ and Cu are shown in Figure 6a. The binding energy of Cu 2p (CSM@ Cu^II^) appeared to have two peaks at 933.88 and 934.48 eV, respectively. These results were attributed to the binding energy of divalent copper ions [45]. The binding energy of Cu 2p also appeared to have two peaks after loading Cu (CSM @ Cu^0^), which were at 932.08 and 934.28 eV, respectively. The results suggest that the copper of the microsphere was present at zero valence after reduction by sodium borohydride. The signal peak of element N (N1s) in XPS is also strong evidence of interactions between the modified chitosan microspheres and copper. Since Cu^2+^ has a free electron orbital, the binding energy of N1s increased, whilst Cu^0^ had a single electron, and the binding energy of N1s decreased. As shown in Figure 6b, the binding energy of N1s in the CSM was 398.88 eV. Compared with the CSM, the binding energy of N1s increased by 1.2 eV after loading Cu^2+^, whilst the binding energy of N1s was reduced by 0.7 eV after loading Cu^0^. The XRD patterns of CSM@ Cu^0^ demonstrate strong Cu^0^ peaks, as shown in Figure 7. There was a diffraction peak at 2θ = 43.33°, corresponding to the diffractive surface of the face-centered cubic (fcc) copper (111) [46].

#### 3.3.4. TEM Characterization

TEM images of the modified chitosan-supported Cu^0^ catalyst (CSM@ Cu^0^) are shown in Figure 8. The nano-copper particles obtained by reducing NaBH_4_ had a relatively narrow particle size distribution between 4 and 12 nm (Appendix A). Through the High Resolution Transmission Electron Microscopy (HR-TEM) test, the lattice spacing of the nano-copper was about 0.21 nm, which is exactly the same as the distance from the (111) crystal plane of the face-centered cubic copper. The percentage of copper in the CSM@ Cu^0^ material was determined to be 3.1% by XPS analysis. Combined with previous analytical tests, we have successfully prepared modified chitosan microspheres loaded with nano-copper.

### 3.4. The Performance of Chitosan Microsphere-Supported Cu Catalyzed in a Boron Addition Reaction.

#### 3.4.1. Procedure of CSM@Cu^0^ Catalyzed in the Boron Addition Reaction

A chitosan-copper catalyst (CSM@Cu^0^), chalcone **4a** (or other derivatives), and bis(pinacolato)diboron were added to a sample vial, and reacted at room temperature for 8 h. After filtration and washing with ethyl acetate, the solvent was evaporated to give *β*-borate compound **5**. Then, compound **5** was added to a mixed solution of tetrahydrofuran- water (THF-H_2_O) containing excess sodium perborate tetrahydrate, and stirred at room temperature for 4 h. After extraction, the target product **6** was isolated by thin column chromatography. The reaction route is illustrated in Figure 9.

#### 3.4.2. Optimization of the Reaction Conditions.

At first, chalcone and bis(pinacol)diboron were used as reactants to study the performance of CSM@ Cu^0^. It should be noted that because the boronated product could be easily decomposed in silica gel, oxidant NaBO_3_·H_2_O was chosen to oxidize the boronate to alcohol to determine the conversion rate. The specific reaction is shown in Table 2. First, the effects of various solvents on the reaction were investigated. In the organic solvent, such as toluene, diethyl ether, acetone, tetrahydrofuran, and so on, the reaction yield of the product **6a** was quite low (<35%, Table 2, Entry 1–4). When methanol was used as the solvent, the yield of the product **6a** could reach 83% (Table 2, Entry 5). Based on our previous findings [37], the yield of product **6a** was greatly increased when an aqueous medium was used instead of a pure solvent, thus MeOH/H_2_O mixture solvent was used. When MeOH /H_2_O = 1:1(*v*:*v*), the yield of product **6a** could be as high as 93% (Table 2, Entry 10). A similar yield could be obtained by using the copper loading capacity from 1 to 5 mol % of the reaction under the first optimization condition. Therefore, 1 mol % copper loading was used to maximize the efficiency. When CSM was used as a catalyst, no boronated product could be detected (Table 2, Entry 17). Additionally, when CSM@Cu^II^ was selected as the catalyst, though a yield of 91% was obtained, serious metal leaching was detected, which would be deadly for catalyst recycling (Table 2, Entry 18). Next, the reaction time was investigated, and almost no improvement could be seen, even when extending the reaction time to 24 h (Table 2, Entry 14–16)**.** Even extending the reaction time to 24 h had almost no effect on the yield. What should be highlighted is that the reaction can be carried out at room temperature in open air smoothly. Therefore, the optimized experimental conditions were 1 mol % CSM@ Cu^0^, an 8 h reaction time, and MeOH/H_2_O (*v*/*v* = 1/1) as the aqueous medium.

### 3.5. Catalyst Reuse

The recyclability of the catalyst was investigated based on the optimal reaction conditions. The solid catalyst was filtered, washed, dried, and added directly to the next cycle. At the same reaction time, the first cycle of the reaction solution was detected by ICP, and copper leaching was not detected. As shown in Figure 10a, although the Cu nanoparticles in the catalyst had a small amount of aggregation after five cycles, the conversion was still as high as 92% (Figure 10b). Therefore, the catalyst had a good stability and could be reused more than five times.

### 3.6. Extension to Other Substrates

Under the optimal reaction conditions, 1 mol % CBS@Cu^0^ catalyst was applied to the boron addition reaction of various a,β-unsaturated acceptors and the results are displayed in Table 3. It can be seen that the chalcone derivatives performed well when they were mono- or di-substituted, such as **6a****–s** except for **6l**. Generally, the electron-withdrawing group lowered the reactivity of the substrate (**6l**, **6q****–r**), whilst the electron-donating group had the opposite enhancing effect (**6g**, **6n**). α,β-unsaturated aliphatic ketone also had a good performance, such as **6t** and **6u** with yields of 83% and 92%, respectively. Unexpectedly, thienyl (**6x**, **6y**) provided unfavorable reaction conditions. In addition, α,β,γ,δ-dienone substrates (**4z**) successfully gave the 1,4-addition product (**6z**) with a low yield of 17%. These results demonstrated that the modified chitosan catalyst supported copper (CBS@Cu^0^) in a heterogeneous catalyst, with an excellent catalytic performance in the boron addition reaction. Both ^1^H NMR and ^13^C NMR of the compounds **6a****–z** are shown in the Appendix A.

## 4. Conclusions

A stable chitosan-based copper catalyst was successfully prepared and supported copper nanoparticles in the microspheres could be well-crystallized. Under optimized experimental conditions, the modified chitosan microsphere-supported Cu catalyst (1 mol %) obtained good catalytic activity in the boridification reaction of aqueous media. The catalyst could be recovered after filtration and maintain its stability following five cycles of repeated use. This work showed that chitosan-supported copper and copper salts are green and highly efficient heterogeneous catalytic materials. Further study on the application of modified chitosan microspheres as organic microporous materials for the removal of corrosive sulfides in transformer oils is warranted.

## Figures and Tables

**Figure 1 polymers-11-01417-f001:**
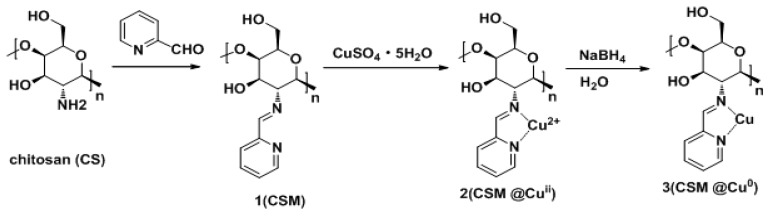
Preparation of chitosan microsphere-supported Cu catalytic materials.

**Figure 2 polymers-11-01417-f002:**
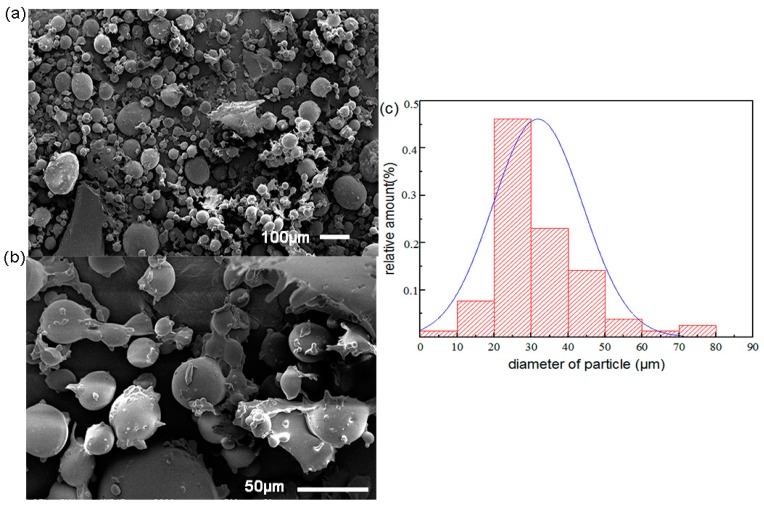
Scanning Electron Microscopy (SEM) analysis of modified chitosan microspheres and their size distribution. (**a**) SEM analysis of modified chitosan microspheres (the scale bar is 100 μm). (**b**) SEM analysis of modified chitosan microspheres (the scale bar is 50 μm). (**c**) Diameter distribution of modified chitosan microspheres.

**Figure 3 polymers-11-01417-f003:**
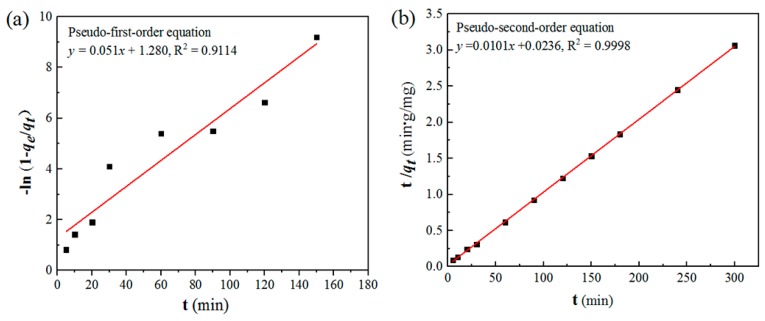
Concentration (100 mg/L) of Cu^2+^ at 25 °C. (**a**) The pseudo-first-order kinetic model fitting results. (**b**) The pseudo-second-order kinetic model fitting results.

**Figure 4 polymers-11-01417-f004:**
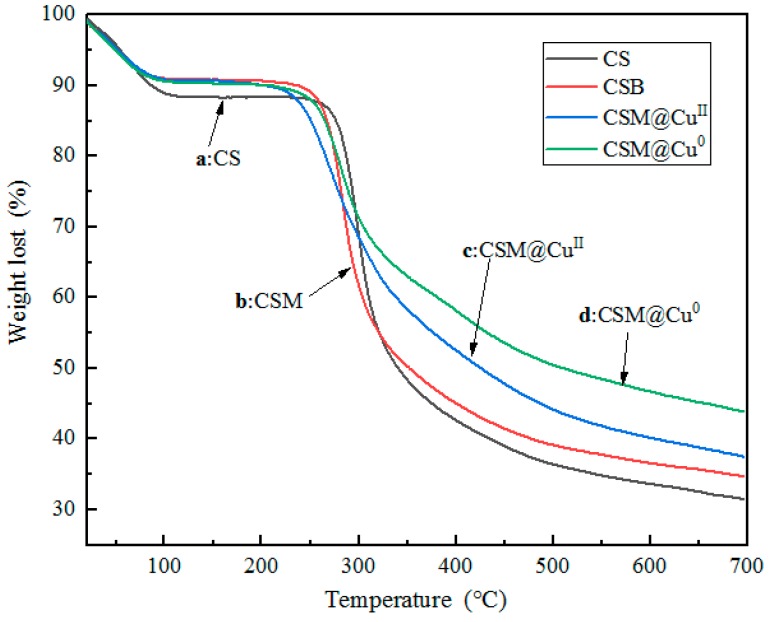
Thermal Gravimetric (TG) curves of the thermal degradation of (**a**) chitosan (CS), (**b**) modified chitosan microspheres (CSM), (**c**) Cu^2+^-loaded modified chitosan microspheres (CSM@ Cu^II^), and (**d**) Cu-loaded modified chitosan microspheres (CSM@Cu^0^).

**Figure 5 polymers-11-01417-f005:**
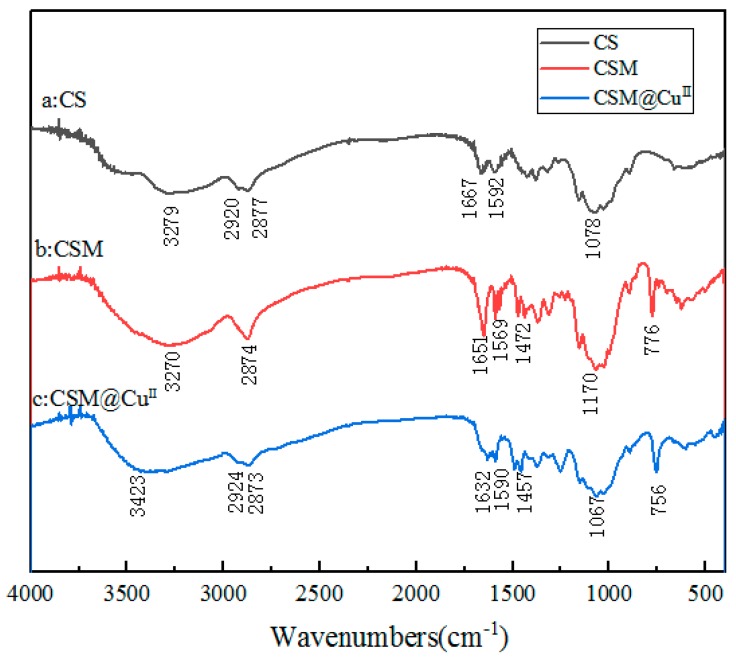
Fourier Transform Infrared Spectroscopy (FTIR) spectra of different compounds: (**a**) chitosan (CS), (**b**) modified chitosan microspheres (CSM), and (**c**) Cu^2+^-loaded modified chitosan microspheres (CSM@ Cu^II^)^.^

**Figure 6 polymers-11-01417-f006:**
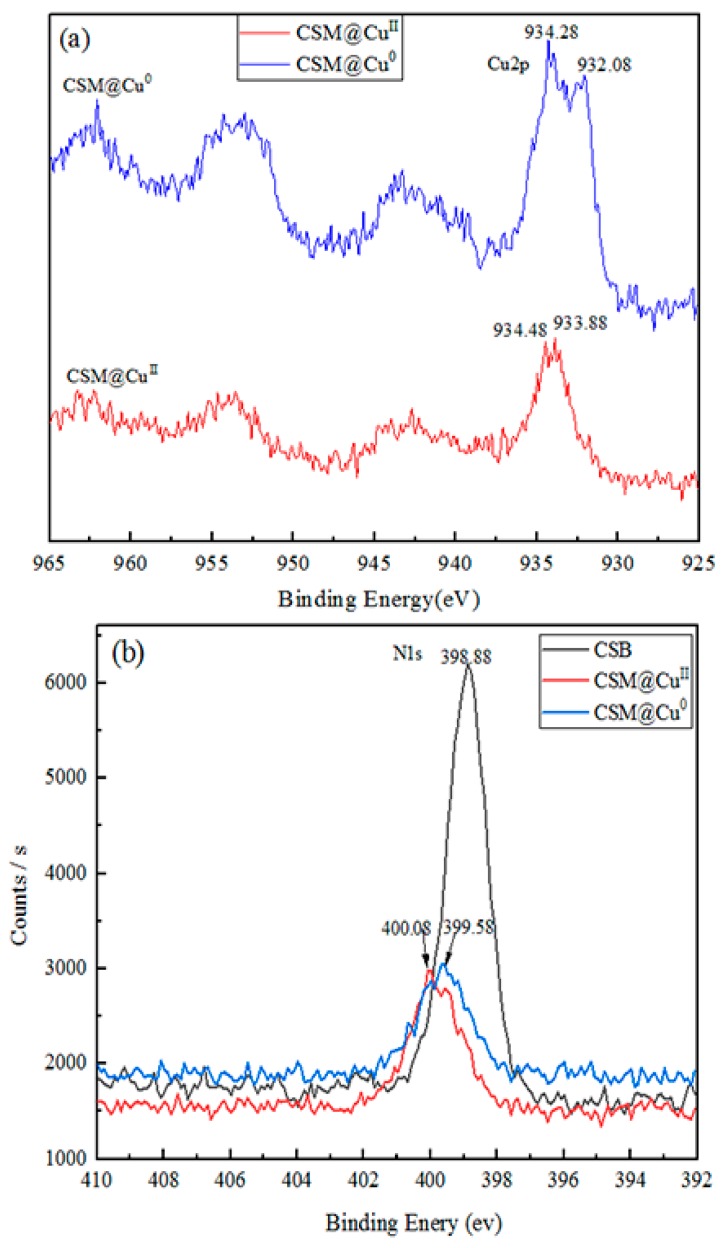
(**a**) X-ray Photoelectron Spectroscopy (XPS) analysis of elemental Cu on Cu^2+^-loaded modified chitosan microspheres (CSM@ Cu^II^) and Cu-loaded modified chitosan microspheres (CSM@ Cu^0^). (**b**) XPS analysis of elemental N on modified chitosan microspheres (CSM), CSM@ Cu^II^, and CSM@ Cu^0^.

**Figure 7 polymers-11-01417-f007:**
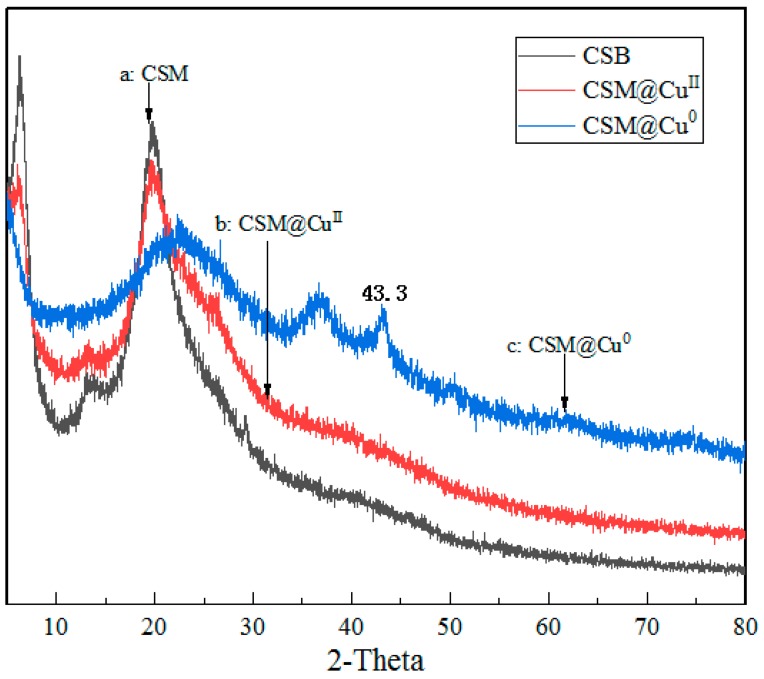
X-ray Diffractometry (XRD) analysis of elemental Cu on (**a**) modified chitosan microspheres (CSM), (**b**) Cu^2+^-loaded modified chitosan microspheres (CSM@ Cu^II^), and (**c**) Cu-loaded modified chitosan microspheres (CSM@ Cu^0^).

**Figure 8 polymers-11-01417-f008:**
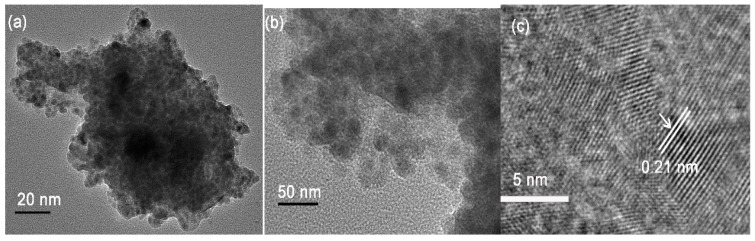
Transmission Electron Microscopy (TEM) analysis of copper nanoparticles on the modified chitosan microsphere. (**a**), (**b**) TEM image of copper nanoparticles. (**c**) TEM analysis of lattice spacing of copper nanoparticles.

**Figure 9 polymers-11-01417-f009:**
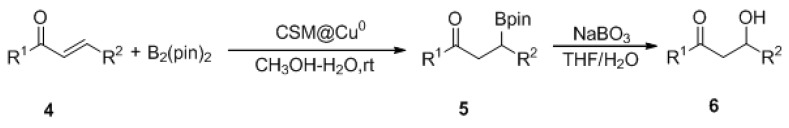
The general route of the boron addition reaction.

**Figure 10 polymers-11-01417-f010:**
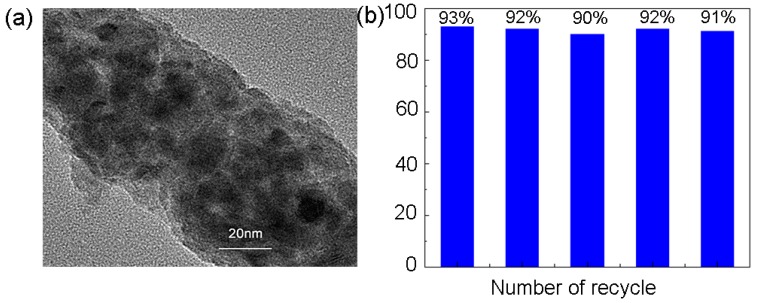
(**a**) Transmission Electron Microscopy (TEM) images of Cu-loaded modified chitosan microspheres (CSM@ Cu^0^) after recycling five times. (**b**) Stability of CSM@ Cu^0^ over the boron addition reaction.

**Table 1 polymers-11-01417-t001:** Dynamic adsorption of copper ions by chitosan microspheres.

Entry	Conc. before Adsorption (*C_0_*, mg/L)	Adsorption Time (min)	Conc. after Adsorption (*Ci*, mg/L)	Adsorption conc. (mg/L)	Adsorption (%)	Absorption Capacity *(q_i_,* mg/g)
1	18.96 ^a^	180	0.49	18.47	97.42	18.47
2	40.75 ^a^	180	0.93	39.82	97.72	39.82
3	59.95 ^a^	180	2.21	57.74	96.31	57.74
4	81.22 ^a^	180	3.11	78.11	96.17	78.11
5	100.63 ^a^	180	2.62	98.01	97.40	98.01
6	100.63 ^a^	5	45.72	54.91	54.57	54.91
7	100.63 ^a^	10	26.33	74.3	73.83	74.3
8	100.63 ^a^	20	17.25	83.38	82.86	83.38
9	100.63 ^a^	30	4.30	96.33	95.73	96.33
/10	100.63 ^a^	60	3.13	97.5	96.89	97.5
11	100.63 ^a^	90	3.09	97.54	96.93	97.54
12	100.63 ^a^	120	2.72	97.81	97.20	97.81
13	100.63 ^a^	150	2.70	97.93	97.32	97.93
14	100.63 ^a^	240	2.67	97.96	97.35	97.96
15	100.63 ^a^	300	2.69	97.94	97.33	97.94
16	100.63 ^b^	180	97.33	3	2.98	3
17	100.63 ^c^	180	64.42	36.19	35.96	36.19
18	100.63 ^d^	180	15.67	84.96	84.43	84.96

^a^ The pH of the solution is 5. ^b^ The pH of the solution is 2. ^c^ The pH of the solution is 3. ^d^ The pH of the solution is 4.

**Table 2 polymers-11-01417-t002:**
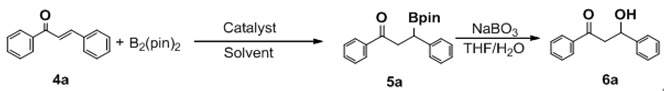
Optimization of the Reaction Conditions.

Entry.	Catalyst	Solvent (1 mL)	Time (h)	Yield (%)
1	CSM@ Cu^0^	Toluene	8	11
2	CSM@ Cu^0^	Et_2_O	8	4
3	CSM@ Cu^0^	Acetone	8	24
4	CSM@ Cu^0^	THF	8	35
5	CSM@ Cu^0^	MeOH	8	83
6	CSM@ Cu^0^	H_2_O	8	26
7	CSM@ Cu^0^	MeOH:H_2_O (4:1)	8	87
8	CSM@ Cu^0^	MeOH:H_2_O (3:1)	8	90
9	CSM@ Cu^0^	MeOH:H_2_O (2:1)	8	89
10	CSM@ Cu^0^	MeOH:H_2_O (1:1)	8	93 ^a^
11	CSM@ Cu^0^	MeOH:H_2_O (1:2)	8	85
12	CSM@ Cu^0^	MeOH:H_2_O (1:3)	8	84
13	CSM@ Cu^0^	MeOH:H_2_O (1:1)	8	85
14	CSM@ Cu^0^	MeOH:H_2_O (1:1)	6	73
15	CSM@ Cu^0^	MeOH:H_2_O (1:1)	12	92
16	CSM@ Cu^0^	MeOH:H_2_O (1:1)	24	93
17	CSM	MeOH:H_2_O (1:1)	8	NR
18	CSM@ Cu^II^	MeOH:H_2_O (1:1)	8	91 ^b^

^a^ Reaction conditions: Chalcone (0.2 mmol), B_2_(pin)_2_ (1.2 eq.), catalyst (0.1 eq.), solvent (1 mL), rt, 8h. ^b^ Metal leaching is 33%.

**Table 3 polymers-11-01417-t003:** Substrate scope of α,β-unsaturated acceptors.

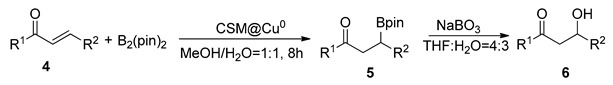
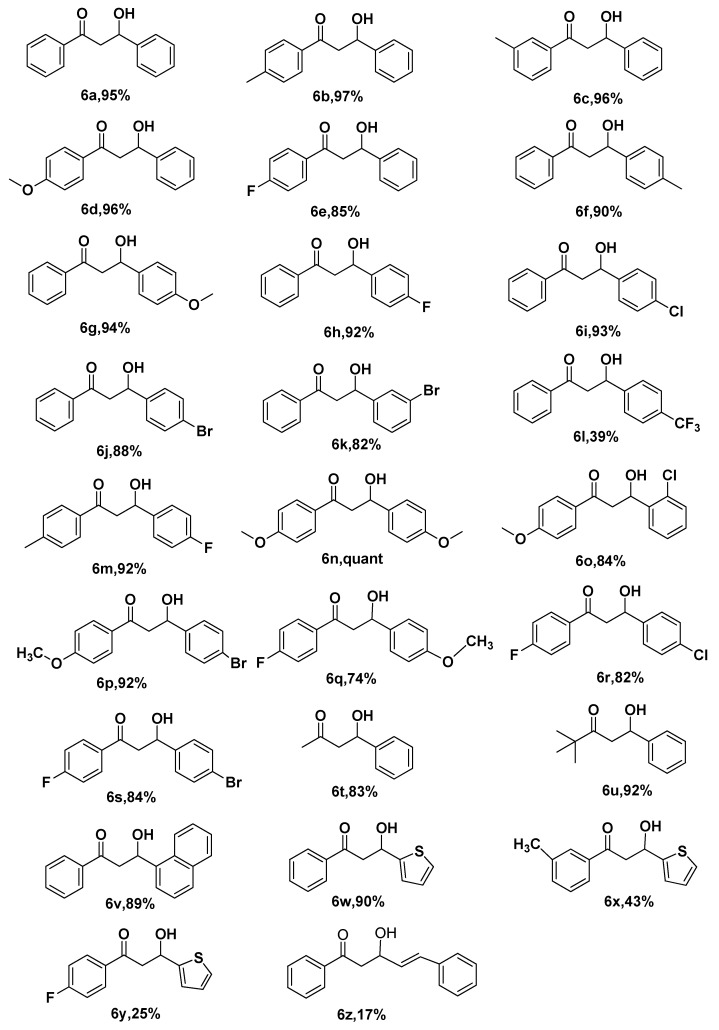

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
