# Peer review of "Preparation of Modified Chitosan Microsphere-Supported Copper Catalysts for the Borylation of α,β-Unsaturated Compounds"

_polymers, 2019, doi:10.3390/polym11091417_

Round 1
Reviewer 1 Report
In this paper, the authors prepared modified-chitosan microspheres by 2-pyridinecarboxaldehyde, and then loaded nano-Cu as catalyst on the as-prepared microspheres by the reduction reaction. The CSM@Cu was successfully applied to catalyze the boronation of unsaturated receptors in water-methanol medium with yields ranging from 17% to 100%. Some questions and comments are given in the following.
1. The authors said the chitosan microspheres were prepared by emulsion crosslinking using 2-pyridinecarboxaldehyde. However, Fig. 1 shows that 2-pyridinecarboxaldehyde had only one aldehyde group and reacted with the amino group on CS. The authors should clearly indicate the crosslinking reaction.
2. By using elemental analyzer, the authors concluded that approximately 5% of the nitrogen sites in the raw materials were acetylated, and approximately 50% of the remaining amino groups had been functionalized with pyridine-2-carbaldehyde. The authors should clearly describe how to obtain these results from the elemental analyzer.
3. It is well known that the chitosan itself can adsorb Cu(II) very well. The authors should compare the adsorptivity of the CS and CSM.
4. The authors said the peak of 1667 cm-1 and 1592 cm-1 belonged to NH2-C=O. This is not correct.
5. After the reduction of Cu (II) to obtain Cu by NaBH4, would the Cu still be chelated very well on the CSM? What would be the adsorptivity of Cu on the chitosan microspheres?
6. There are some typo and syntax errors.
Author Response
Point 1. The authors said the chitosan microspheres were prepared by emulsion crosslinking using 2-pyridinecarboxaldehyde. However, Fig. 1 shows that 2-pyridinecarboxaldehyde had only one aldehyde group and reacted with the amino group on CS. The authors should clearly indicate the crosslinking reaction.
Response 1: Thank you for your suggestion. The chitosan microspheres were prepared by emulsion methods, we have deleted “crosslinking” and changed using other expression.Related sentences are as follows:“Chitosan microspheres modified by 2-pyridinecarboxaldehyde were prepared” and “We prepared 2-pyridineformaldehyde modified chitosan microspheres (CSM) by emulsion reaction,”
Point 2. By using elemental analyzer, the authors concluded that approximately 5% of the nitrogen sites in the raw materials were acetylated, and approximately 50% of the remaining amino groups had been functionalized with pyridine-2-carbaldehyde. The authors should clearly describe how to obtain these results from the elemental analyzer.
Response 2: Thank you for your constructive suggestion. We have added the elemental content to the support material, see Table 1S. Molecular formula of chitosan is (C6H11NO4)n, molecular formula of schiff base chitosan is (C12H14N2O4)n, and average molecular formula of modified chitosan microspheres (CSM) is (C9.6H12.6N1.47O4)n. By estimation, approximately 53% of the remaining amino groups had been functionalized with pyridine-2-carbaldehyde
Table 1S. The content of N, C, H, O of chitosan microspheres (CS) and modified chitosan microspheres (CSM)
|
Compound |
H(w%) |
C(w%) |
N(w%) |
O(w%) |
C:N:O(Atomic%:Atomic%:Atomic%) |
|
|
CS |
6.81 |
44.91 |
8.63 |
39.65 |
6.04:0.994:4 |
|
|
CSM |
5.93 |
54.23 |
9.72 |
30.12 |
9.6: 1.473:4 |
|
Point 3. It is well known that the chitosan itself can adsorb Cu(II) very well. The authors should compare the adsorptivity of the CS and CSM.
Response 3: Thank you for your constructive suggestion. We have added the adsorptivity of the CS and CS to the support material, see Table S2
Point 4. The authors said the peak of 1667 cm-1 and 1592 cm-1 belonged to NH2-C=O. This is not correct.
Response 4: Thank you for your suggestion. We have revised this sentence to: “the peak of 1667 cm-1 belonged to C=O (NH2-C=O), the peak of 1592 cm -1 belonged to C-N and N-H (NH2-C=O)”. Related reference books are “Xiaowen Shi, Hongbing Deng, Yumin Du, Chitin/chitosan material and its application [M], 2015, p 59-62, ISBN:978-7-122-24155-9”.
Point 5. After the reduction of Cu (II) to obtain Cu by NaBH4, would the Cu still be chelated very well on the CSM? What would be the adsorptivity of Cu on the chitosan microspheres?
Response 5: Thank you for your suggestion. After the reduction of Cu (II) to obtain Cu, some black powder could be seen came off from the microspheres, but it would be absorbed again in centrifugation and performed very well even after recycled five times. As shown in Fig. 6b, compared with CSM, the binding energy of N1s was reduced by 0.7 eV after loading Cu0, which means the Cu still be chelated very well on the CSM.
Point 6. There are some typo and syntax errors.
Response 6: Thanks a lot for the kindly suggestions. Based on your helpful comments, the language usage has been revised by a native English-speaker engaged through the auspices of a profession a proof reading service.

Reviewer 2 Report
General comments:
The characterization of the catalyst (sections 3.1, 3.2 and 3.3) seems to be entirely reliable, but its generalized application to the borylation of "any" α,β-unsaturated compound (as the title implies) is not supported. The fact that the content of the section on borylation has been broken down and dispersed into subsections 2.5, 3.4 and 3.6, together with the insufficient characterization of the compounds listed in Table 3, lends itself to being interpreted as an evasive maneuver to overcome the difficulty to achieve the ambitious objective of the work. The authors have exceeded themselves in the generalization of the application of their results (demonstrated only for β-hydroxydihydrochalcone derivatives), and this can only be corrected by modulating the interpretation made. In its current form, doubts may be cast on the credibility of the article's achievements.
Specific comments:
Please be consistent in the use of the term "borylation" It is strongly advised to modify the title: rephrasing to "...borylation of β-hydroxydihydrochalcone derivatives" would be much more accurate. The generalization intended by the authors is exaggerated (and should be avoided all throughout the ms.) Line 52: add a space between "as" and "nucleophile" Line 61: the letter size of "[36]" is smaller than the one used for the text in the template. Please increase it. Line 66: explain what CSM is upon first usage (chitosan microspheres) Lines 79, 80 and 81: add a space before mL, g/mL and mL, respectively. Line 87: add a space before mg. Section 2.5 is not clear and reference 25 does not seem very appropriate, because it is difficult to recognize a Sonogashira coupling reaction. Hence, section 2.5 must be entirely redone, and it should be integrated with sections 3.4 and 3.6. The discussion needs to be improved.
Author Response
Point 1. Please be consistent in the use of the term "borylation" It is strongly advised to modify the title: rephrasing to "...borylation of β-hydroxydihydrochalcone derivatives" would be much more accurate.
Response 1: Thanks a lot for the kindly suggestions. β-hydroxydihydrochalcone derivatives were product, not a raw material for borylation, so we think the title should be retained.
Point 2. The generalization intended by the authors is exaggerated (and should be avoided all throughout the ms.)
Response 2: Thank you for your suggestion. We have deleted the sentence: “Our data demonstrated that chitosan microspheres provide a sustainable route for the synthesis of organoboron compounds.”
Point 3. Line 52: add a space between "as" and "nucleophile"
Response 3: Thank you for your suggestion. A space has been added between "as" and "nucleophile".
Point 4. Line 61: the letter size of "[36]" is smaller than the one used for the text in the template. Please increase it.
Response 4: Thank you for your suggestion. The letter size of "[36]" has been increased to normal text.
Point 5.Line 66: explain what CSM is upon first usage (chitosan microspheres)
Response 5: Thank you for your suggestion. We added CSM when we first used it.
Point 6. Lines 79, 80 and 81: add a space before mL, g/mL and mL, respectively. Line 87: add a space before mg.
Response 6: Thank you for your suggestion. Space has been added to the reminded and other needed location.
Point 7. Section 2.5 is not clear and reference 25 does not seem very appropriate, because it is difficult to recognize a Sonogashira coupling reaction. Hence, section 2.5 must be entirely redone, and it should be integrated with sections 3.4 and 3.6. The discussion needs to be improved.
Response 7: Thank you for your suggestion.I’m sincerely sorry, the reference 25 in the title “2.5 Procedure of CSM@Cu0 catalyzed in boron addition reaction 25” is a mistake which has been deleted. Besides, Section 2.5 has been moved to section 3 to integrated with sections 3.4 and 3.6.

Round 2
Reviewer 1 Report
As chitosan was not crosslinked, would the modified CS microspheres be swelling or even soluble in acidic solutions? The swelling capability in acidic solutions at different pH values becomes very important which would affect the adsorption capability of copper ion. For chitosan with high DDA, the peak of 1667 cm-1 belonged to amide I (C=O stretching in NH2-C=O) and the peak of 1592 cm -1 belonged to the amino group (-NH2 bending). In Table S2, the note "a" should be CSM and also the pH value should be given.
Author Response
Point 1. As chitosan was not crosslinked, would the modified CS microspheres be swelling or even soluble in acidic solutions? The swelling capability in acidic solutions at different pH values becomes very important which would affect the adsorption capability of copper ion.
Response 1: Thanks a lot for the kindly suggestions. I agree with your point of view. As the pH of the solution decreases, the swelling ability of chitosan increases significantly (see table below), and the chitosan were partially dissolved in the acidic solution. We have added the sentence to: ‘This phenomenon may also be caused by the swelling ability of CMS at different pH values, which affects the adsorption capacity for copper ions.”
In the swelling capability experiments, microspheres 1 (0.1g) were treated with 20 mL hydrochloric acid solution and vibrated at a rate of 220 r/min at 25 °C. After 3 hours, the microspheres were taken out and the surface water was wiped off with filter paper, and then quickly weighed. Finally,the microspheres were dried at 100 °C.
Table 3S. Swelling capability of CSM at different pH value
|
Entry |
pH |
Initial quality (m0, g) |
Quality after swelling for 3h (mi, g) |
Quality after drying (mf, g) |
Swelling capability (SC= ) |
|
1 |
2 |
0.1036 |
0.2766 |
0.0812 |
240.6 |
|
2 |
3 |
0.1070 |
0.2480 |
0.0939 |
164.1 |
|
3 |
4 |
0.1047 |
0.1528 |
0.0965 |
58.3 |
|
4 |
5 |
0.0999 |
0.1465 |
0.0977 |
49.9 |
Point 2. For chitosan with high DDA, the peak of 1667 cm-1 belonged to amide I (C=O stretching in NH2-C=O) and the peak of 1592 cm -1 belonged to the amino group (-NH2 bending).
Response 1: Thank you for your suggestion. We have revised this sentence to: “the peak of 1667 cm-1 belonged to amide I (C=O stretching in NH2-C=O), the peak of 1592 cm -1 belonged to N-H (-NH2 bending).”
Point 3. In Table S2, the note "a" should be CSM and also the pH value should be given.
Response 3: I’m sincerely sorry for this mistake. We have revised the note to: “a. CSM, pH =5. b. CS, pH =5.”